# Longitudinal Assessment of Multimorbidity Medication Patterns among Smokers in the COPDGene Cohort

**DOI:** 10.3390/medicina59050976

**Published:** 2023-05-18

**Authors:** Yisha Li, Sarah J. Schmiege, Heather Anderson, Nicole E. Richmond, Kendra A. Young, John E. Hokanson, Stephen I. Rennard, Tessa L. Crume, Erin Austin, Katherine A. Pratte, Rebecca Conway, Gregory L. Kinney

**Affiliations:** 1Department of Epidemiology, University of Colorado Anschutz Medical Campus, Aurora, CO 80045, USA; yisha.li@cuanschutz.edu (Y.L.);; 2Department of Biostatistics and Informatics, University of Colorado Anschutz Medical Campus, Aurora, CO 80045, USA; 3Department of Clinical Pharmacy, University of Colorado Anschutz Medical Campus, Aurora, CO 80045, USA; 4Division of Pulmonary, Critical Care and Sleep Medicine, University of Nebraska Medical Center, Omaha, NE 68198, USA; 5Mathematical and Statistical Sciences, University of Colorado Denver, Denver, CO 80204, USA; 6Division of Biostatistics and Bioinformatics, National Jewish Health, Denver, CO 80206, USA

**Keywords:** chronic diseases, medication patterns, latent class analysis, smoker

## Abstract

*Background and objectives*: Chronic obstructive pulmonary disease (COPD) is usually comorbid with other chronic diseases. We aimed to assess the multimorbidity medication patterns and explore if the patterns are similar for phase 1 (P1) and 5-year follow-up phase 2 (P2) in the COPDGene cohort. Materials and Methods: A total of 5564 out of 10,198 smokers from the COPDGene cohort who completed 2 visits, P1 and P2 visits, with complete medication use history were included in the study. We conducted latent class analysis (LCA) among the 27 categories of chronic disease medications, excluding COPD treatments and cancer medications at P1 and P2 separately. The best number of LCA classes was determined through both statistical fit and interpretation of the patterns. *Results*: We found four classes of medication patterns at both phases. LCA showed that both phases shared similar characteristics in their medication patterns: LC0: low medication; LC1: hypertension (HTN) or cardiovascular disease (CVD)+high cholesterol (Hychol) medication predominant; LC2: HTN/CVD+type 2 diabetes (T2D) +Hychol medication predominant; LC3: Hychol medication predominant. *Conclusions*: We found similar multimorbidity medication patterns among smokers at P1 and P2 in the COPDGene cohort, which provides an understanding of how multimorbidity medication clustered and how different chronic diseases combine in smokers.

## 1. Introduction

Cigarette smoking is a primary causative factor for a multitude of chronic diseases, including chronic obstructive pulmonary disease (COPD) [1,2]. In addition to COPD, people who smoke frequently have multimorbidity (the presence of multiple diseases or conditions), including heart disease, cancers and diabetes. Comorbidities increase the risk of poorer quality of life, higher disability life years lost, and earlier age at death [1]. Despite the high prevalence of comorbidities among those with COPD, longitudinal multimorbidity patterns are not well-understood [1,2]. In a cohort at high risk of COPD, typically observed morbidities include cardiovascular disease (CVD), diabetes, hyperlipidemia, and obesity [2]. Identifying diseases by diagnosis code is commonly used; however, this specificity is not always accessible in observational studies. In a previous study, we applied a novel approach using medication information as an indication of multimorbidity among smokers at high risk of COPD [3]. Previously, we identified 8 unique medication patterns among a total of 10,127 former and current smokers at phase (P1) in the COPDGene cohort [3]. These eight medication patterns independently predicted mortality and acute COPD exacerbations in the COPDGene cohort [3]. Since this study reported medication patterns at P1, a longitudinal assessment of medications of both P1 and phase 2 (P2) will be useful to describe medication patterns as an indication of multimorbidity patterns over time.

The aim of this study is to assess the prevalence and stability of multimorbidity treatment patterns at two study visits after accounting for P1 mortality and loss of follow-up in the COPDGene cohort. Our work is to provide a better understanding of both morbidity treatment and morbidity patterns to advance clinical insights into how different chronic diseases combine and whether multimorbidity medication (MM)patterns remain stable among smokers at high risk of COPD.

## 2. Materials and Methods

COPDGene is a multicenter observational study designed to identify genetic factors associated with COPD [4]. COPDGene enrolled 10,198 former and current smokers aged 45–80 years with a history of ≥10 pack-year smoking with and without COPD from initiation in P1 from 21 clinical centers across the United States (Figure 1) [4]. All participants were given consent before entering the study. COPDGene started P1 enrollment in 2008 and finished P1 enrollment in 2011, and five-year follow-up data were completed in P2 [5]. We used the COPDGene^®^ to study P1 and P2 data. A total of 5564 participants who had both P1 and P2 medication and demographic information were included in our study cohort (Figure 1). Medication information was collected through self-report and confirmed through study personnel evaluation of participants’ bottles of medications, including both prescribed and over the count medication in each phase of their study visit. Medication exposure status was reported as “yes/no” to indicate if a specific medication was taken. Medication groups were further categorized based on expert opinion [3]. A total of 27 categories of medications were identified in both phases (Appendix A. A list of self-reported morbidities in the COPDGene was published in previous literature [2]. The self-reported morbidities are dichotomous (yes/no) variables reported separately from medication use, which reflects whether participants self-reported having specific diseases.

Latent class analysis (LCA) using the 27 categories of medications was conducted to assess patterns of medication use. Identification of the best number of latent classes of medication patterns is achieved by testing multiple a priori number of classes and evaluating different model fits using goodness-of-fit indices. We initially evaluated two classes and increased the classes sequentially. According to the literature, there is some agreement that to select a final LCA model, (1) multiple fit statistics should be used or reported [6]; (2) the Bayesian information criterion (BIC) may be the most reliable fit statistic and should routinely be reported [6,7,8]; and (3) reasonable interpretability should be considered in choosing solution [6,7,9,10,11]. We included the comparison of Akaike information criterion (AIC), BIC, sample size adjusted BIC (SABIC), and entropy. The best model was finally determined by BIC, the likelihood-ratio test (LRT), and the overall interpretability of the model [12,13]. A lower value of AIC, BIC, and SABIC indicates a better model fit, and a higher value of entropy (≥0.8) shows increased classification precision [14,15]. After determining the optimal classes of LCA, all participants were classified into the class where they had the highest predicted probability of class membership, and the best class of membership for each individual was automatically selected through LCA. The label for each LCA class was determined by the observed item-response probabilities and labeled by names of dominating categories of medications. Item-response probability is considered high if the response probability is >0.7 and low if the response probability is <0.3. The naming of each latent class (LC) is based on the heterogeneity of medication categories in each LC, the names of predominant medication categories, and how the characteristics of each LC are distinguished from others [3,16,17]. We repeated the same LCA methods using the P2 data.

Latent transitional analysis (LTA), a longitudinal version of LCA, tested the longitudinal measurement invariance (likelihood-ratio test, LRT: *p* > 0.05) of item-response probabilities between P1 and P2 [18]. If the longitudinal measurement invariance in the LTA cannot be achieved (LRT, *p* < 0.05), the phase-specific LCA is used to report patterns over time. Either the LTA or the phase-specific LCA provided us with results of how individuals moved from one pattern to a different pattern or remained in the same pattern in each phase. We summarized the movement of medication patterns among individuals between phases to illustrate how MM patterns changed over time.

MM patterns with the participants’ self-reported comorbidities in each phase were compared through logistic regression to determine the utility of medication patterns serving as a proxy for diagnosis. 

All statistical analyses were performed using SAS Version 9.4 (SAS Institute Inc.: Cary, NC, USA), and a *p*-value < 0.05 is considered statistically significant. LCA figures were created through the R package ggplot2, version 3.6.1 [19].

## 3. Results

The average age of participants was 60 years in P1; 50% of the participants were female, and 28% of participants identified as Non-Hispanic Black or African American (Table 1). A total of 47% of the participants reported being current smokers in P1, and the rest were former smokers (Table 1). The average body mass index (BMI) was 29 in P1.

The comparison of LCA fit statistics of the 2-class solution through the 8-class solution based on 27 classes of medications in P1 and P2 separately is shown in Table 2. BIC selected the 5-class solution and the 6-class solution in P1 and P2, respectively; however, the results of LCA were not interpretable after the 4-class solution in either phase. The BIC in the 3-class solution and 4-class solution were the same in P1, but an LRT showed the 4-class solution significantly improved model fit compared to the 3-class solution (G^2^Δ:241.39, dfΔ:28, *p* < 0.0001). The 5-class solution has the same BIC as the 6-class solution in P2, but the entropy is only 0.66 (Table 2). In addition, the 5-class solution model did not show as distinguished characteristics as the 4-class solution model in P2. Therefore, a 4-class solution model was also selected for P2.

LTA was applied for the 4-class solution models for both phases. However, we did not achieve measurement invariance because the non-constrained model showed a significantly better fit through LRT in both phases (LRT: *p* < 0.0001, Appendix A). Therefore, results of the medication patterns are reported for the 4-class solution LCA model for each phase separately.

### 3.1. Characteristics of LCA Patterns at P1

The first of the four latent classes for P1 (denoted LC0_p1) included 3138 (56.40%) participants. Participants in the LC0_p1 had a low probability of all medication use (probability of each medication <6%, Figure 2), and the medications used in this group did not share the same clustering with other LCA classes. LC0_p1 was labeled as “Low medication”.

The second of the four latent classes for P1 (LC1_p1) included 690 (12.40%) participants. LC1_p1 had a high probability of angiotensin-converting-enzyme inhibitors (ACEi, 97.68%) use and a moderate probability of antiplatelet (40.52%), diuretic (41.31%) and statin (55.81%) use (Figure 2). Both ACEis and diuretics are medications primarily treating hypertension (HTN) [20]. Antiplatelets primarily decrease the risk of stroke, heart attack or other CVD [21]. Statins primarily reduce low-density lipoprotein and treat high cholesterol (Hychol) [22,23]. LC1_p1 was labeled as “HTN/CVD+ Hychol predominant”.

The third of the four latent classes for P1 (LC2_p1) included 255 (4.6%) participants. LC2_p1 had a high probability of biguanides (72.44%) and statin (73.20%) use, moderate probability of ACEi (45.45%), antiplatelet (46.50%) and sulfonylureas (40.34%) use (Figure 2). Biguanides and sulfonylureas are primarily used to treat T2D [24,25]. LC2_p1 was labeled as “HTN/CVD+T2D+Hychol predominant”.

The last latent classes for P1 (LC3_p1) included 1481 (26.6%) participants. LC3_p1 had a moderate probability of statin (47.21%) use (Figure 2). LC3_p1 was labeled as “Hychol predominant”.

### 3.2. Characteristics of LCA Patterns at P2

The first of the four latent classes for P2 (LC0_p2) included 2386 (42.90%) participants. LC0_p2 had a low probability of all medication use (probability of each medication <6%, Figure 2), and the medications used in this group did not share the same clustering with other LCA models at P2. LC0_p2 was labeled as “Low medication”.

The second of the four latent classes for P2 (LC1_p2) included 485 (8.70%) participants. LC1_p2 had a moderate probability of ACEi (41.15%) and antiplatelet (66.70%) (Figure 2). LC1_p2 had a high probability of beta blocker (98.17%) and statin (79.06%) use (Figure 2). ACEi and Beta blocker are medications that primarily treat HTN or CVD [20,26]. LC1_p2 was labeled as “HTN/CVD+Hychol predominant”.

The third of the four latent classes for P2 (LC2_p2) included 450 (8.10%) participants. LC2_p2 had a moderate probability of ACEi (51.69%) and antiplatelet (40.46%) use (Figure 2). LC2_p2 had a high probability of biguanides (74.11%) and statin (69.83%) use (Figure 2). LC2_p2 was labeled as “HTN/CVD+T2D+ Hychol predominant”.

The last latent class for P2 (LC3_p2) included 2,243 (40.30%) participants. LC3_p2 had a moderate probability of Statin (50.43%) use (Figure 2). LC3_p2 was labeled as “Hychol predominant”.

The numbers of individuals that changed medication patterns from P1 to P2 are shown in Table 3. Most participants remained in the same patterns in LC0 (low medication, 63.64%), LC2 (HTN/CVD + T2D + Hychol predominant, 52.55%) and LC3 (Hychol predominant, 62.19%) from P1 to P2. A total of 53.62% of participants in LC1 (HTN/CVD + Hychol predominant) in P1 transitioned to LC3 (Hychol predominant) in P2 (Table 3).

MM patterns are consistent with their primarily treated self-report comorbidity patterns (Appendix A). There was a total of 1148 participants that died in the 5-year follow-up (P1), and 1134 of them had baseline medication information. We included their P1 medication patterns profile and death rate based on our previous published outcomes (Appendix A) [3].

Figure 2 shows P1 and P2 medication patterns among participants who completed both phases: red color shows a higher probability (close to 1), and yellow color shows a lower probability (close to 0) of having a specific category of medication. X-axis includes labels of each medication pattern in phase 1 (left side) and phase 2 (right side) separately. Labels are explained as follows: LC0_p1 had a low probability of all medication use (probability of each medication <6%). LC1_p1 selected angiotensin-converting-enzyme inhibitors (ACEi, 97.68%), antiplatelet (40.52%), diuretics (41.31%) and statin (55.81%). LC2_p1 selected biguanides (72.44%), statin (73.20%), ACEi (45.45%), antiplatelet (46.50%) and sulfonylureas (40.34%). LC3_p1 selected statin (47.21%). LC0_p2 had a low probability of all medication use (probability of each medication <6%. LC1_p2 selected ACEi (41.15%), antiplatelet (66.70%), beta blocker (98.17%) and statin (79.06%). LC2_p2 selected ACEi (51.69%), antiplatelet (40.46%), biguanides (74.11%) and statin (69.83%). LC3_p2 selected statin (50.43%).

## 4. Discussion

We found consistent MM patterns based on a longitudinal assessment of 5564 current or former smokers who completed both P1 and P2 in the COPDGene cohort (Figure 2). The importance of looking at both P1 and P2 medication is to inform healthcare givers regarding the patterns of multiple medication use and long-term chronic conditions among smokers. Our work is innovative because our longitudinal medication patterns demonstrated that some chronic diseases progress concurrently, and our treatment patterns provide insights into examining and treating human diseases as an integrated paradigm. Our study has several strengths. To our knowledge, our study is the first study exploring longitudinal MM treatment patterns in the COPDGene cohort. We also applied an LTA approach to test whether the patterns at P1 and P2 are concordant from a statistical perspective. We identified the same four latent classes representing patterns of medication use at P1 and P2 in the COPDGene cohort: LC0: low medication, LC1: HTN/CVD + Hychol predominant, LC2: HTN/CVD + T2D + Hychol predominant, LC3: Hychol predominant. Similar medication patterns indicated that this cohort has consistent chronic diseases of HTN, CVD, T2D and hyperlipidemia. This is consistent with the report of consequences in smokers that heart disease, diabetes, stroke, and COPD are the most prevalent diseases among smokers in the past 50 years [1].

We did not include COPD medication in our analysis because participants’ pulmonary function was evaluated through spirometry and CT scan and can be identified through the global initiative of COPD status in Table 1. Even though the medication patterns showed similarity, we did not achieve measurement invariance in LTA. The reason we did not achieve measurement invariance is that some of the parameter estimates of the medication category were not the same in P1 and P2. For example, LC2_p1 and LC2_p2 were both labeled as HTN/CVD + T2D + Hychol medication; however, LC1_p1 selected both biguanides and sulfonylureas as potential T2D treatment, but LC2_p2 only selected biguanides. We found consistent MM patterns at a population level in the COPDGene cohort in both P1 and P2; however, the MM structure may not remain the same in each phase at an individual level. Individual treatment patterns could fluctuate over time because of changes in the number of diseases or changes in treatments or treatment regimens. 

Our study showed there were changes in medication patterns at the individual level between groups from P1 to P2 (Table 3). Most participants in LC0, LC2 and LC3 remained in the same LCA class (Table 3). LC1 group had the most transition (53.62%) from LC1_p1 (HTN/CVD + Hychol predominant) to LC3_p2 (Hychol predominant) (Table 3). It is possible these people still had HTN or CVD in P2 and transitioned to less prevalent HTN/CVD treatment due to changes in their prescriptions, which are not captured by LCA because of the low frequency of use. It is also possible that some people started to manage HTN/CVD through lifestyle modification such as cessation of smoking, changing diet or exercise [27]. The numbers of medication patterns discovered in this work were different from our preliminary findings of 8 medication patterns in earlier published work, but this result was from data based on a total of 10,127 participants in P1 [3]. Subsequently, the difference is because our current study was based on 5564 participants (survivors of the 10,127 participants) who completed both phases, so there will be some changes in both numbers and characteristics of the medication patterns contributed either due to mortality or loss of follow-up. The differences in demographics in P1 among participants in our study cohort (completed both P1 and P2) and participants excluded are described in Appendix A. 

Even though the numbers of medication patterns were different from our previous study, we were still capturing most of the morbidities we found previously, including HTN, T2D, CVD and hyperlipidemia [3]. There was increased use of beta blocker in LC1 at P2 compared to P1 (Figure 2). This might be explained by the recommendations from both the American Heart Association and European Society of Cardiology that beta blocker and ACEi should be applied to all people who had heart failure with reduced ejection fraction to prevent symptomatic heart failure and reduce hospitalization rate and mortality [28,29,30]. Beta blocker was also a first-line treatment recommended for people who coexist with atrial fibrillation to control ventricular rate [28,29,30].

Our study had several limitations. First, there were some new medications included in P2 for diabetic treatment, which were not available for P1 because they were not introduced as a new treatment available in the market between the two phases. For example, Glucagon-like peptide 1 inhibitors were included in other diabetic medications in P2 but not in P1. However, this difference did not impact our study outcome because only two participants used this medication in P2, and other diabetic medications were not selected in the LCA model in either phase. Second, we did not collect information regarding patients’ medication dosage and adherence; thus, we cannot confirm whether participants used a specific medication. It is also likely to have report bias because patients might forget to bring their medication bottles to their visits which results in over-reporting or under-reporting their medication. Third, as we discussed in our previous paper, the same medication can be applied to treat different diseases, but we only make inferences about the primary diseases treated by medications [3]. For example, we cannot determine whether people were using HTN medications to treat chronic kidney disease (CKD) or HTN. The interaction between CKD and HTN is complex, but resistant HTN is commonly seen in patients with CKD [31]. Fourth, our LCA model indicates medication patterns but does not reflect every single medication used by each individual. However, the probability of using each category of medication within each pattern can be found in our LCA item-response probability. Fifth, it is possible that participants received more novel therapeutic interventions, lifestyle adjustments or over-the-counter medications to treat their diseases that were not included in our model because the information is not available in our data. However, it is an informative approach to capture medication patterns as a proxy of multimorbidity at a population level. Sixth, our study did not use validated comorbidity evaluation tools like the Charlson comorbidity index (CCI) score to confirm our approach because we do not have access to the diagnosis code for CCI calculation [32]; however, the predominant diseases captured through our approach were consistent with previously published literature [1]. Understanding medication patterns is also a useful tool for making inferences about disease patterns through treatment decisions when accurate diagnosis information is absent. Seventh, our study did not compare the difference in medication use or medication patterns between current and former smokers, which will be addressed in our future work. Lastly, our study is limited by survivor bias because we only included participants who completed both P1 and P2 with 5-year survival. Therefore, the generalization of these MM patterns applies to relatively healthier smokers. Future studies should reproduce MM patterns using other cohorts with smokers.

## 5. Conclusions

We found consistent MM patterns in each phase among 5564 participants who completed both P1 and P2 in the COPDGene study. Despite known changes in medication use guidelines and transitions of treatments, our approach showed there were similar MM patterns in smokers in two consecutive phases. Because chronic diseases are not curable even after being treated, these MM patterns are indications that captured consistent chronic disease patterns of HTN, CVD, T2D and hyperlipidemia among current and former smokers at a population level. Our finding provides an understanding of both morbidity treatment and morbidity patterns to advance clinical insights into how different chronic diseases combine in smokers.

## Figures and Tables

**Figure 1 medicina-59-00976-f001:**
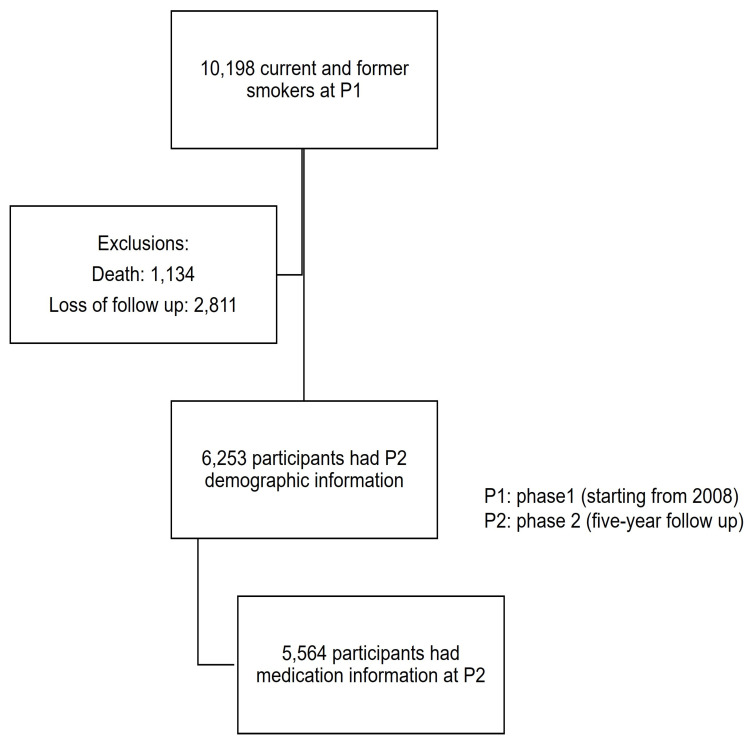
Inclusion of study cohort: 10,127 out of 10,198 smokers had P1 medication data. A total of 1134 out of 10,127 participants died before P2, and 2811 participants had a loss of follow-up at P2. A total of 6235 participants had P2 demographic information, and 5564 participants had P2 medication information.

**Figure 2 medicina-59-00976-f002:**
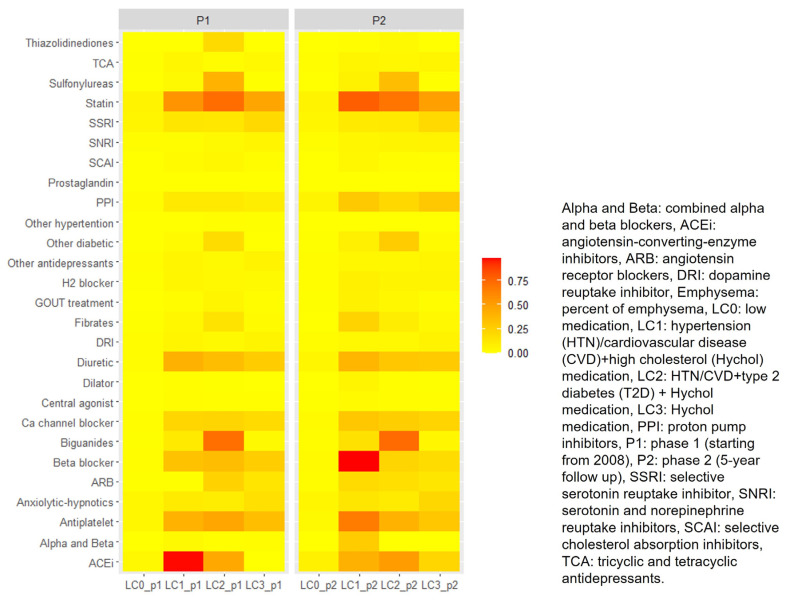
Medication patterns based on 27 classes of medications at P1 and P2.

**Table 1 medicina-59-00976-t001:** Demographics of participants who completed both P1 and P2.

		P1 (*n* = 5564)		P2 (*n* = 5564)	
		n (%)	Mean (SD)	n (%)	Mean (SD)
Age			60.09 (8.68)		65.65 (8.66)
Female		2780 (49.96)		2780 (49.96)	
African American		1584 (28.47)		1584 (28.47)	
Smoking status	Former smoker	2957 (53.15)		3479 (62.53)	
	Current smoker	2607 (46.85)		2085 (37.62)	
Smoking pack-years			42.89 (23.62)		44.35 (23.99)
BMI			29.09 (6.11)		29.00 (6.39)
COPD GOLD	PRISm	689 (12.38)		665 (12.43)	
	GOLD 0	2541 (45.67)		2265 (42.32)	
	GOLD 1	492 (8.84)		511 (9.55)	
	GOLD 2	1122 (20.17)		1092 (20.40)	
	GOLD 3	570 (10.24)		575 (10.74)	
	GOLD 4	150 (2.70)		244 (4.56)	

BMI: body mass index, COPD GOLD: global initiative for chronic obstructive pulmonary disease, P1: phase 1, P2: phase 2 (5-year follow-up), PRISm: preserved ratio impaired spirometry, SD: standard deviation. The total number is not 5564 because of missing data.

**Table 2 medicina-59-00976-t002:** Latent class analysis statistics of medication patterns in P1 and P2.

	G^2^	AIC	BIC	CAIC	ABIC	Entropy
P1						
2-solution	7501.19	7611.19	7975.51	8030.51	7800.74	0.69
3-solution	6936.08	7102.08	7651.88	7734.88	7388.13	0.70
4-solution	6694.69	6916.69	7651.86	7762.86	7299.14	0.76
5-solution	6360.60	6638.60	7559.34	7698.34	7117.64	0.71
6-solution	6219.14	6553.14	7659.36	7826.36	7128.68	0.66
7-solution	6116.41	6506.41	7798.10	7993.10	7178.45	0.66
8-solution	5985.84	6431.84	7909.00	8132.00	7200.38	0.71
P2						
2-solution	11,622.08	11,732.08	12,096.40	12,151.40	11,921.63	0.68
3-solution	11,027.35	11,193.35	11,743.15	11,826.15	11,479.40	0.57
4-solution	10,600.47	10,822.47	11,557.74	11,668.74	11,205.02	0.69
5-solution	10,232.21	10,510.21	11,430.96	11,569.96	10,989.26	0.66
6-solution	9988.95	10,322.95	11,429.17	11,596.17	10,898.50	0.70
7-solution	9826.11	10,216.11	11,507.81	11,702.81	10,888.16	0.73
8-solution	9677.31	10,123.31	11,600.47	11,823.47	10,891.85	0.70

AIC: Akaike Information Criterion, BIC: Bayesian information criterion, SABIC: sample size adjusted BIC, CAIC: consistent AIC, P1: phase 1, P2: phase 2 (5-year follow-up).

**Table 3 medicina-59-00976-t003:** Numbers of individuals in each medication patterns in P1 and P2.

	P2 Low Medication (*n* = 2386)	P2 HTN/CVD + Hychol Predominant (*n* = 485)	P2 HTN/CVD + T2D + Hychol Predominant (*n* = 450)	P2 Hychol Predominant (*n* = 2243)
P1 Low medication (n = 3138)	1997 (63.64)	95 (3.03)	133 (4.24)	913 (29.09)
P1 HTN/CVD + Hychol predominant (n = 690)	80 (11.59)	140 (20.29)	100 (14.49)	370 (53.62)
P1 HTN/CVD + T2D + Hychol predominant (n = 255)	24 (9.41)	58 (22.75)	134 (52.55)	39 (15.29)
P1 Hychol predominant (n = 1481)	285 (19.24)	192 (12.96)	83 (5.60)	921 (62.19)

LC: latent class, P1: phase 1, P2: phase 2 (5-year follow-up), HTN: hypertension, CVD: cardiovascular disease, Hychol: high cholesterol, T2D: type 2 diabetes.

## Data Availability

Data supporting this manuscript is available to all COPDGene investigators. Data are available upon request through https://dccweb.njhealth.org/sec/COPDGene/MainPage.cfm (1 May 2023).

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
