# Peer review of "Longitudinal Assessment of Multimorbidity Medication Patterns among Smokers in the COPDGene Cohort"

_medicina, 2023, doi:10.3390/medicina59050976_

Round 1

Reviewer 1 Report

In this manuscript the author's assess the prevalence of multi morbidity treatment patterns in smokers at a higher risk of COPD. The analysis are well done and the the manuscript is well written and easy to understand. the only advice is to correct some references errors in the results and discussion sections.

Author Response

Thanks for your suggestions. We have corrected the format in the references. 

Reviewer 2 Report

The study conducted by the authors has a good rationale and background data from their previous study. The study design is well planned and executed. A few suggestions to improve the paper are
1. Avoid repetitive sentences in the introduction that does not give new information 

2. The tables and figures need to be referenced in the text that are currently missing.

3. A discussion describing the novelty of the study and why it is important to study both p1 and p2 needs to be highlighted.

Author Response

1. Avoid repetitive sentences in the introduction that does not give new information  

Answer:

Thanks for your comment. We removed some sentences in the introduction. 

2. The tables and figures need to be referenced in the text that are currently missing. 

Answer:

Thanks for your suggestions. We have added references of tables and figures in our text. 

3. A discussion describing the novelty of the study and why it is important to study both p1 and p2 needs to be highlighted. 

Answer:

Thanks for the comment. We have added the following sentences in the discussion:

The importance of looking at both P1 and P2 medication is to inform healthcare givers regarding the patterns of multiple medication use and long-term chronic conditions among smokers. Our work is innovative because our longitudinal medication patterns demonstrated that some chronic diseases progress concurrently, our treatment patterns provide insights into examining and treating human diseases as an integrated paradigm”

Reviewer 3 Report

The article “Longitudinal assessment of multimorbidity medication patterns among smokers in the COPDGene cohort” addresses a very important topic. I have following comments/suggestions,

1.       The abstract and introduction sections are very well written.

2.       In method section or elsewhere, I cant see the details for ethical or IRB approval number.

3.       Some information is required regarding the participants of the study. How were they selected? Was the informed consent taken?

4.       The figure 2, the x-axis is hard to follow.

5.       Reference No 22, please correct the style.

Author Response

  1. The abstract and introduction sections are very well written. 

Answer:

Thanks for the comment.

  1. In method section or elsewhere, I can’t see the details for ethical or IRB approval number.

Answer:

Thanks for the comment. We have added the IRB approval number in acknowledgement. Details of IRB was included in the acknowledgements as follows: 

“We thank COPDGene investigators who helped acquiring the data for this study. COPDGene is supported by Award Number U01 HL089897 and Award Number U01 HL089856 from the National Heart, Lung, and Blood Institute. The central institutional reviewer board (IRB) number for COPDGene phase 1, phase 2 and phase 3 is HS-2778. COPDGene is also supported by the COPD Foundation through contributions made to an Industry Advisory Board that has included AstraZeneca, Bayer Pharmaceuticals, Boehringer-Ingelheim, Genentech, GlaxoSmithKline, Novartis, Pfizer, and Sunovion.”

  1. Some information is required regarding the participants of the study. How were they selected? Was the informed consent taken? 

Answer:

Thanks for your comment.  We have included the following sentences in methods and materials: 

“COPDGene is a multicenter observational study designed to identify genetic factors associated with COPD.[4] COPDGene enrolled 10,198 former and current smokers, aged 45-80 years with a history of ≥10 pack-year smoking with and without COPD from initiation at P1 from 21 clinical centers across the United States (Figure 1).[4] All participants were given consent before entering the study. COPDGene started P1 enrollment from 2008 and finished P1 enrollment in 2011, and five-year follow up data was completed in P2.[5] We used the COPDGene® study P1 and P2 data. A total of 5,564 participants who had both P1 and P2 medication and demographic information were included in our study cohort (Figure 1).”

  1. The figure 2, the x-axis is hard to follow. 

Answer:

Thanks for your comment. We have added the following sentence explaining the x-axis in figure 2.

“X-axis includes labels of each medication pattern at phase 1 (left side) and phase 2 (right side) separately. Labels are explained as follows: LC0_p1 had a low probability of all medication use (probability of each medication <6%). LC1_p1 selected angiotensin-converting-enzyme inhibitors (ACEi, 97.68%), antiplatelet (40.52%), diuretic (41.31%), and statin (55.81%). LC2_p1 selected biguanides (72.44%), statin (73.20%), ACEi (45.45%), antiplatelet (46.50%), and sulfonylureas (40.34%). LC3_p1 selected statin (47.21%). LC0_p2 had a low probability of all medication use (probability of each medication <6%. LC1_p2 selected ACEi (41.15%), antiplatelet (66.70%), beta blocker (98.17%) and statin (79.06%). LC2_p2 selected ACEi (51.69%), antiplatelet (40.46%), biguanides (74.11%) and statin (69.83%). LC3_p2 selected statin (50.43%).”

  1. Reference No 22, please correct the style. 

Answer:

Thanks for the comment. We have corrected reference No 22. 

Reviewer 4 Report

This is a very well-written manuscript. The data is comprehensive and informative. One general question, in your study, you included current and former smokers with COPD at P1 and P2 and then classified the medication patterns in all COPD. Have you investigated the difference in the patterns between current and former smoking? Will former smokers have fewer medications? 

Author Response

Have you investigated the difference in the patterns between current and former smoking? Will former smokers have fewer medications?  

Answer:

Thanks for your comment. We included the discussion of this part in our limitations:

Seventh, our study did not compare the difference of medication use or medication patterns between current and former smokers, which will be addressed in our future work”